# Long-Term TalkingFace Generation
# via Motion-Prior Conditional Diffusion Model

**Fei Shen** [1]  **Cong Wang** [2]  **Junyao Gao** [3]  **Qin Guo** [4]  **Jisheng Dang** [5]  **Jinhui Tang** [1]  **Tat-Seng Chua** [6]

## Abstract

Recent advances in conditional diffusion models have shown promise for generating realistic TalkingFace videos, yet challenges persist in achieving consistent head movement, synchronized facial expressions, and accurate lip synchronization over extended generations. To address these, we introduce the **M**otion-priors **C**onditional **D**iffusion **M**odel (**MCDM**), which utilizes both archived and current clip motion priors to enhance motion prediction and ensure temporal consistency. The model consists of three key elements: (1) an archived-clip motion-prior that incorporates historical frames and a reference frame to preserve identity and context; (2) a present-clip motion-prior diffusion model that captures multimodal causality for accurate predictions of head movements, lip sync, and expressions; and (3) a memory-efficient temporal attention mechanism that mitigates error accumulation by dynamically storing and updating motion features. We also introduce the TalkingFace-Wild dataset, a multilingual collection of over 200 hours of footage across 10 languages. Experimental results demonstrate the effectiveness of MCDM in maintaining identity and motion continuity for long-term TalkingFace generation.

## 1. Introduction

TalkingFace generation (Tan et al., 2024; Peng et al., 2024; Ye et al., 2024; Ji et al., 2021; Tan et al., 2023; Kim et al., 2018; Liang et al., 2022; Ye et al., 2023; Pumarola et al., 2018; Vougioukas et al., 2020) aims to create realistic and expressive videos from a reference face and audio, with applications in virtual avatars, gaming, and filmmaking (Shen

[1]Nanjing University of Science and Technology [2]Nanjing University [3]Tongji University [4]Peking University [5]Sun Yat-sen University [6]National University of Singapore. Correspondence to: Jinhui Tang <jinhuitang@njust.edu.cn>.

*Proceedings of the $42^{nd}$ International Conference on Machine Learning*, Vancouver, Canada. PMLR 267, 2025. Copyright 2025 by the author(s).

et al., 2023). However, the complexity of facial movements, including head, lip, and expression motions, presents challenges, along with the need to maintain identity consistency across extended sequences.

Early methods (Vougioukas et al., 2020; Wang et al., 2021b; Hong et al., 2022a; Chan et al., 2022; Guo et al., 2024) use GANs (Goodfellow et al., 2014; Mirza & Osindero, 2014) to synthesize facial motions onto a reference image through a two-step process: decoupling motion features from audio and mapping them onto intermediate representations like facial landmarks (Yang et al., 2023), 3DMM (Sun et al., 2023), or HeadNeRF (Hong et al., 2022b). Despite their promise, GAN-based methods suffer from training instability and inaccuracies in motion extraction, often leading to artifacts like blurriness and flickering that compromise video realism. Recent diffusion models (Wei et al., 2024; Shen et al., 2025a; Tian et al., 2024; Shen et al., 2025b;d; Guo et al., 2024; Zheng et al., 2024; Jiang et al., 2024) have improved TalkingFace generation by enhancing video realism through multi-step denoising that preserves conditional input information. These methods typically use a Reference UNet (Hu, 2024) to encode identity features and integrate audio via cross-attention. However, reliance on static audio features and weak correlations between audio and motion complicate the decoupling of identity and motion cues, often resulting in artifacts like motion distortion and flickering, especially in long-term generation.

While some methods (Wang et al., 2024b; Ma et al., 2024; Yang et al., 2024) improve long-term stability by introducing motion constraints like facial landmarks and emotion tags, these constraints often overly bind poses to the reference image, limiting expression diversity. Models trained with driven landmark fail to learn natural audio-driven motion patterns, reducing audio-visual synergy. Additionally, static emotion tags cannot capture dynamic shifts, leading to rigid, inauthentic animations over extended sequences. Besides, some approaches (Xu et al., 2024; Chen et al., 2024) inject brief motion reference frames, usually fewer than five over 0.2 seconds, which is insufficient to establish coherent motion, resulting in random, less dynamic movements.

In this paper, we propose the **M**otion-priors **C**onditional **D**iffusion **M**odel (**MCDM**) to address the challenges in

achieving long-term consistency in TalkingFace generation. The MCDM comprises three key modules: the archived-clip motion-prior, the present-clip motion-prior diffusion model, and a memory-efficient temporal attention mechanism. Unlike conventional reference UNet-based identity learning, the archived-clip motion-prior introduces historical frames along with a reference frame via frame-aligned attention , enhancing identity representation and creating a cohesive facial context over extended sequences. Then, the present-clip motion-prior diffusion model leverages multimodal causality and temporal interactions to effectively decouple and predict motion states, including head, lip, and expression movements, ensuring a clear separation between identity and motion features and promoting temporal consistency across frames. To support long-term stability, we devise a memory-efficient temporal attention that dynamically stores and updates historical motion features, integrating them with current motion cues via a memory update mechanism. This structure reduces error accumulation often observed in diffusion-based long-term TalkingFace generation, enabling more stable and consistent outputs. Additionally, we present the TalkingFace-Wild dataset, a high-quality, multilingual video dataset with over 200 hours of footage in 10 languages, offering a valuable resource for further research in Talking-Face generation. Our main contributions are summarized as follows:

- We propose MCDM to enhance robust identity consistency and support temporal consistency in long-term TalkingFace generation.

- MCDM leverages archived-clip priors for identity-aware context, present-clip priors for disentangling identity and motion, and memory-efficient temporal attention to integrate historical and current motion features with reduced error accumulation.

- MCDM achieves state-of-the-art performance on multiple TalkingFace benchmarks, demonstrating superior identity preservation and temporal consistency under long-term generation.

## 2. Related Work

**GAN-Based Methods.** GAN-based approaches (Kim et al., 2018; Zhou et al., 2020; Pumarola et al., 2018; Vougioukas et al., 2020; Zhang et al., 2023; Wang et al., 2021b; Hong et al., 2022a; Chan et al., 2022; Guo et al., 2024) for TalkingFace generation extract motion features from audio or visual inputs and map them to intermediate representations such as facial landmarks (Yang et al., 2023), 3DMM (Sun et al., 2023), or HeadNeRF (Hong et al., 2022b). MakeItTalk (Zhou et al., 2020) employs LSTMs to predict landmarks from audio, followed by a warp-based

GAN for video synthesis. GANimation (Pumarola et al., 2018) models facial motion via continuous manifolds, enhancing expression dynamics. SadTalker (Zhang et al., 2023) integrates ExpNet and PoseVAE to refine motion representations within the FaceVid2Vid (Wang et al., 2021b) framework. DaGAN (Hong et al., 2022a) introduces self-supervised geometric learning to capture dense 3D motion fields. While effective, GAN-based methods suffer from adversarial training instability and motion inaccuracies, often resulting in artifacts that degrade realism.

**Diffusion-Based Methods.** Diffusion models (Rombach et al., 2022; Shen & Tang, 2024; Wang et al., 2024a; Shen et al., 2025c) have gained traction in TalkingFace generation, producing high-quality, diverse outputs. AniPortrait (Wei et al., 2024) maps audio to 3D facial structures, generating temporally coherent videos with expressive detail. MegActor-$\Sigma$ (Wang et al., 2024b) synchronizes lip movements, expressions, and head poses using a reference UNet (Hu, 2024) and facial loss functions to enhance fidelity. Hallo (Xu et al., 2024) and EchoMimic (Chen et al., 2024) leverage limited motion reference frames to improve expression diversity and pose alignment. However, reliance on short-term frame histories (2-4 frames) compromises long-term motion consistency, while increased frame dependencies escalate computational costs. Additionally, static audio features and restricted references fail to capture natural motion variations, leading to artifacts such as motion distortion and rigid expressions in extended sequences.

Unlike prior work, our approach introduces motion priors from both archived and present clips to enhance long-term motion prediction and identity consistency. By leveraging historical frames and memory-efficient temporal attention, MCDM improves motion continuity while maintaining realism in TalkingFace generation.

## 3. Method

**Task Definition.** Given a reference image, audio, and optional facial landmarks, TalkingFace generation aims to produce temporally coherent and realistic videos. The key challenges include maintaining consistent identity over time, achieving natural head movements, and ensuring expressive and precise lip alignment with audio cues. However, existing methods often encounter limitations such as error accumulation, inconsistent identity preservation, suboptimal audio-lip synchronization, and rigid expressions.

### 3.1. Overall Framework

To address the above challenges, we introduce MCDM, a framework centered on a denoising UNet resembling Stable Diffusion v1.5 (SD v1.5) [1], tailored to denoise multi-frame

---

[1]https://huggingface.co/runwayml/stable-diffusion-v1-5

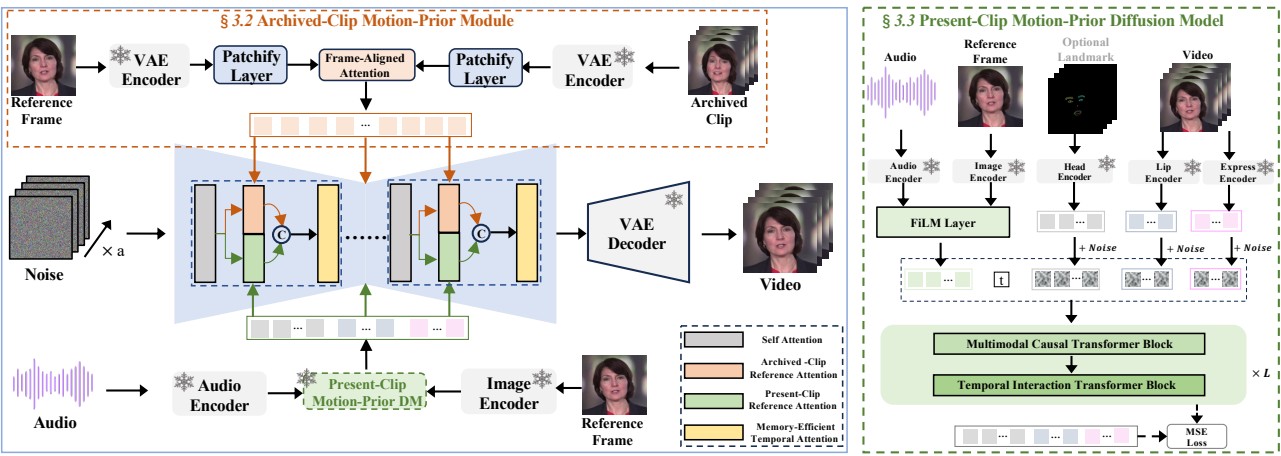

Figure 1. **Our MCDM architecture.** On the upper, the archived-clip motion-prior leverages frame-aligned attention with archived-clip, enhancing identity coherence over extended sequences. On the right, the present-clip motion-prior diffusion model uses multimodal causality and temporal interactions to decouple and predict motion states, covering head, lip, and expression movements while maintaining a clear separation of identity and motion features.

noisy latent inputs under conditional guidance. As illustrated in Figure 1, unlike standard UNet architectures, each Transformer block in MCDM incorporates four attention layers. The first layer, a self-attention, mirrors that in SD v1.5. The second and third layers are parallel cross attention (spatial-wise), designed for distinct interactions: the archived-clip reference attention layer, which integrates motion priors from archived clip encoded by the archived-clip motion-prior module (Section 3.2), and the present-clip reference attention, which engages with present clip priors from the present-clip motion-prior diffusion model (Section 3.3). The fourth layer, the memory-efficient temporal attention (Section 3.4), is a temporal-wise self attention that dynamically updates and merges archived motion features with current motion features, effectively mitigating error accumulation.

### 3.2. Archived-Clip Motion-Prior Module

**Motivation.** Existing methods typically use the past $2 - 4$ frames to guide the denoising network for generating temporally consistent videos. However, this limited history frame is insufficient for maintaining long-term consistency, and incorporating more frames exponentially increases computational demand, making it impractical for real-world applications. To overcome these limitations, we propose an archived-clip motion prior that integrates long-term historical frames and a reference frame into the denoising UNet via conditional frame-aligned attention, enhancing identity representation and establishing motion context.

**Architecture.** As illustrated in Figure 1, the archived-clip motion-prior consists of two frozen VAE encoders, two learnable patchify layers, and a frame-aligned attention mechanism. Given a reference frame $X_{\text{ref}} \in \mathbb{R}^{b \times 1 \times c \times h \times w}$

and a archived clip $X_{\text{arch}} \in \mathbb{R}^{b \times a \times c \times h \times w}$, where $b$, $c$, $h$, $w$, and $a$ represent the batch size, channels, height, width, and the number of archived frames, respectively. First, the frozen VAE encoder extracts latent features from both the reference and archived frames, resulting in $f_x \in \mathbb{R}^{b \times 1 \times 4 \times \frac{h}{8} \times \frac{w}{8}}$ and $f_a \in \mathbb{R}^{b \times a \times 4 \times \frac{h}{8} \times \frac{w}{8}}$, respectively. Next, the learnable patchify layers, consisting of 2D convolutions followed by flattening operations, transform these latent features into tokens, yielding $F_x \in \mathbb{R}^{b \times 1 \times m \times d}$ and $F_a \in \mathbb{R}^{b \times a \times m \times d}$, where $m$ and $d$ denote the token length and embedding dimension.

In the frame-aligned attention, we adopt a frame-wise computation approach to improve efficiency and adaptability for long temporal sequences. For each archived frame $i \in [1, a]$, the Key $K_i$ is derived from the reference tokens $F_x$, while the Value $V_i$ is derived from the tokens of the corresponding archived frame $F_a^i$:

$$K_i = F_x \mathbf{W}_K, \quad V_i = F_{a_i} \mathbf{W}_V, \quad (1)$$

where $\mathbf{W}_K \in \mathbb{R}^{d \times d}$ and $\mathbf{W}_V \in \mathbb{R}^{d \times d}$ are learnable projection matrices for the Key and Value. The attention for each frame $i$ is then computed as:

$$\text{Attention}(Q, K_i, V_i) = \text{Softmax}\left(\frac{QK_i^\top}{\sqrt{d}}\right)V_i, \quad (2)$$

where $Q \in \mathbb{R}^{n \times d}$ represents a learnable query tokens, with $n$ denoting the number of queries. Aggregating the outputs across all frames yields the final output $F_{ac} \in \mathbb{R}^{b \times a \times n \times d}$, where each frame's attended tokens reflect both the static reference and dynamic temporal information.

### 3.3. Present-Clip Motion-Prior Diffusion Model

**Motivation.** Motion information is typically driven either by landmark signals from a driving video or directly by

audio cues. The landmark-driven approach guides reference image movements but limits the natural diversity of head motions and expressions. In contrast, audio-driven methods rely solely on audio cues, often lacking sufficient guidance for realistic head movement. To address these limitations, we propose the present-clip motion-prior diffusion model, which first predicts motion states, including head, lip, and expressions motions, rather than directly generating TalkingFace videos.

**Architecture.** We aim to predict motion in head, lip, and expressions lip movements, conditioned on audio and image tokens. As shown in Figure 1 (right), we begin by extracting feature tokens from the audio encoder, image encoder, head encoder, lip encoder, and express encoder.

**Audio Encoder:** Audio sequence tokens are extracted from the input audio via a frozen Wav2Vec model (Baevski et al., 2020).

**Image Encoder:** Image tokens are extracted from the reference frame using a frozen CLIP (Radford et al., 2021) and are replicated along the temporal dimension to align with audio features.

**Head Encoder:** Head tokens are extracted from reference landmark video through a frozen Landmark Guider [2]; notably, these tokens are optional, allowing simulation of conditions with or without reference video guidance.

**Lip and Express Encoders:** Lip and expression tokens are extracted from the target video using a custom-trained encoder. Details of the lip and express encoders are provided in the supplementary material.

We then pass the audio and image tokens through a feature-wise linear modulation (FiLM) layer (Perez et al., 2018) to adaptively learn multimodal correlation tokens. These tokens, along with the timestep $t$, and noise-added tokens for head, lip, and expression movements, are prepended to the input sequence. This composite input is fed into an $L$-layer structure consisting of a multimodal causal transformer block (Peebles & Xie, 2023) and a temporal interaction transformer block (Hu, 2024), with added noise in facial motion tokens acting as the supervision. The training loss $L_{\text{prior}}$ for the present-clip motion-prior diffusion model $\epsilon_\theta$ is defined as:

$$L_{\text{prior}} = \mathbb{E}_{t,F_p,z_t,\epsilon}, \|\epsilon - \epsilon_\theta\left(z_t, t, F_p\right)\|^2. \quad (3)$$

Without landmark guidance, $F_p$ represent multimodal interaction tokens from audio and the reference frame. $z_t$ represent noise-added tokens for head, lip, and expression movements at timestep $t$. With landmark guidance, $F_p$ additionally include landmark tokens. $z_t$ represent noise-added lip and expression tokens. This design allows flexible con-

[2]https://github.com/MooreThreads/Moore-AnimateAnyone

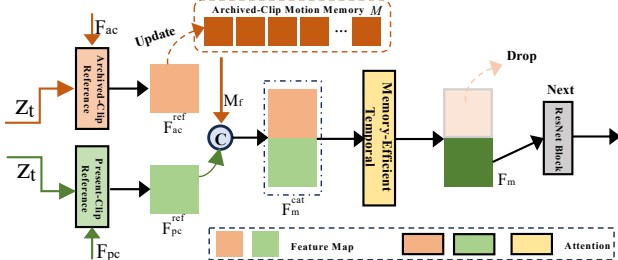

*Figure 2.* **The overview of memory-efficient temporal attention.** It can dynamically update and integrate historical motion features with current ones.

ditioning, incorporating landmark guidance when available, while effectively leveraging multimodal interactions for accurate motion state predictions.

### 3.4. Memory-Efficient Temporal Attention

**Motivation.** For long-term TalkingFace generation, current methods primarily adopt either fully or semi-autoregressive strategies: the former generates one frame per iteration, while the latter produces a fixed-length clip. However, due to GPU memory limitations, relying on a restricted frame history for extrapolation often results in error accumulation, as limited prior motion information undermines consistency over extended sequences. Therefore, we propose a memory-efficient temporal attention to dynamically update and integrate historical motion features with current ones, reducing error accumulation.

**Architecture.** AnimateDiff (Guo et al., 2023) demonstrates that the temporal layer in self-attention ensures smooth temporal continuity and consistency of appearance details across frames. We replace traditional self-attention with fast attention (Choromanski et al., 2020) in the temporal layer to enhance temporal continuity and manage memory efficiently, allowing the accumulation of extensive historical motion information for consistent long-sequence generation. As shown in Figure 2, let $F_{ac}$ and $F_{pc}$ denote the output features of the archived-clip and present-clip motion-prior modules, respectively, and let $Z_t$ represent the noisy latent feature at time step $t$. These features undergo reference attention, yielding refined representations $F_{ac}^{\text{ref}}$ and $F_{pc}^{\text{ref}}$, which capture spatial-domain motion characteristics. $F_{ac}^{\text{ref}}$ is then input into the motion memory update mechanism, which aggregates motion across frames, producing the update feature $M_f$. The memory update mechanism is defined as follows,

**(1) Initialization**: At the first frame, the memory $M_1$ is initialized with $F_{ac}^{\text{ref}}$ since no prior motion information exists:

$$M_1 = F_{ac}^{\text{ref}}. \quad (4)$$

**(2) Memory Update**: For each frame $f$, the memory $M_f$ is updated by combining the current feature $F_{ac}^{\text{ref}}$ with the

previous memory $M_{f-1}$ as:

$$M_f = \alpha M_{f-1} + (1-\alpha)F_{ac}^{\text{ref}}, \qquad (5)$$

where $\alpha \in [0, 1]$ controls the balance between past and current frames. This fixed memory update mechanism avoids storage bottlenecks of historical information. We then concatenate $F_{pc}^{\text{ref}}$ with $M_f$ along the temporal dimension, creating $F_m^{\text{cat}}$, which integrates past and current motion. $F_m^{\text{cat}}$ is processed through Fast Attention along the temporal axis to capture dependencies across frames, with the lower half of the resulting feature map used as the output $F_m$.

### 3.5. Training and Inference

**Training.** Our training process is divided into three stages, each with specific learning objectives. Each stage is supervised using standard MSE loss (Rombach et al., 2022).

**Stage1.** The archived-clip motion-prior is trained to enhance identity representation and establish a robust facial motion context across extended sequences. The present-clip reference attention and memory-efficient temporal attention modules remain frozen during this stage.

**Stage2.** The present-clip motion-prior diffusion model is trained to predict the motion states of facial expressions, lip, and head movements. To simulate scenarios without a driving video, we randomly drop the entire landmark clip.

**Stage3.** The full motion-priors conditional diffusion model is trained for generating stable and consistent long-term TalkingFace videos. Only the present-clip reference and memory-efficient temporal attentions are trained.

**Inference.** The present-clip motion-prior diffusion model first predicts distinct motion tokens based on the given conditions (either with or without landmark guidance). Landmarks are not used by default unless specified. Subsequently, MCDM utilizes these motion tokens, alongside a single reference image and audio input, to generate the video sequence. For the initial archived clip, we initialize it using the reference image and then progressively update the motion memory to ensure temporal consistency.

## 4. Experiments

### 4.1. Experimental Settings

**Datasets.** The HDTF dataset (Zhang et al., 2021) comprises 410 videos with over 10,000 unique speech sentences, varied head poses, and movement patterns. Following prior work (Chen et al., 2024; Tian et al., 2024; Xu et al., 2024), we split HDTF into training and testing sets with a 9:1 ratio. The CelebV-HQ dataset (Zhu et al., 2022) includes 35,666 clips (3–20 seconds each) across 15,653 identities, totaling roughly 65 hours. Both datasets present quality issues, such as audio-lip misalignment, facial occlusions, small facial

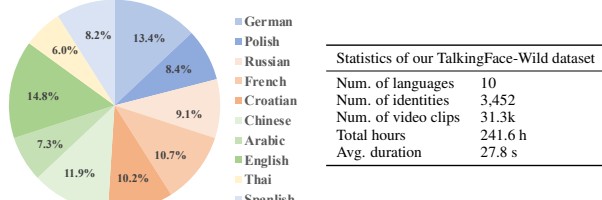

| Statistics of our TalkingFace-Wild dataset | |
|---|---|
| Num. of languages | 10 |
| Num. of identities | 3,452 |
| Num. of video clips | 31.3k |
| Total hours | 241.6 h |
| Avg. duration | 27.8 s |

*Table 1.* **Statistics of our TalkingFace-Wild dataset.** We release a TalkingFace dataset that is well-balanced across 10 languages.

regions, and low resolution. To mitigate these, we developed a custom data processing pipeline for high-quality TalkingFace data, detailed in the following subsection.

Additionally, mostly methods (Wang et al., 2024b; Xu et al., 2024; Jiang et al., 2024) employ proprietary datasets for supplementary training and testing. Similarly, we sourced a variety of TalkingFace videos from YouTube using targeted keyword queries (e.g., "nationality," "interview," "dialogue") across different languages and contexts. From Table 1, we collect a new high-quality dataset, **TalkingFace-Wild**, covering 10 languages and totaling over 200 hours after processing through our data pipeline. To assess the generalization capability of models, we also constructed an open-set test collection of 20 diverse portrait images and 20 audio clips.

**Data Processing.** First, we detect scene transitions in raw videos using PySceneDetect[3] and trim each clip to a maximum duration of 30 seconds. Next, we apply face detection (Guo et al., 2021) to exclude videos lacking complete faces or containing multiple faces, using the bounding boxes to extract talking head regions. Third, an image quality assessment model (Su et al., 2020) filters out low-quality and low-resolution clips. Fourth, SyncNet (Prajwal et al., 2020) assesses audio-lip synchronization, discarding clips with misaligned audio. Finally, we manually inspect a subset to verify audio-lip synchronization and overall video quality, ensuring precise filtering. In addition, to ensure a fair comparison, we report results trained independently on each of the previously mentioned datasets.

**Metrics.** We utilize a comprehensive set of metrics to assess the quality of generated videos and audio-lip synchronization. Fréchet Inception Distance (FID) (Heusel et al., 2017) evaluates individual frame quality by comparing feature distributions from a pre-trained model. Fréchet Video Distance (FVD) (Unterthiner et al., 2019) quantifies the distributional distance between real and generated videos, providing an overall assessment of video fidelity. Sync-C and Sync-D (Chung & Zisserman, 2017) evaluate lip synchronization from content and dynamic perspectives, with higher Sync-C and lower Sync-D scores indicating superior alignment with audio. Structural Similarity Index (SSIM) (Wang et al.,

---

[3] https://github.com/Breakthrough/PySceneDetect

| Method | HDTF | | | | | | CelebV-HQ | | | | | |
|---|---|---|---|---|---|---|---|---|---|---|---|---|
| | FID↓ | FVD↓ | Sync-C↑ | Sync-D↓ | SSIM↑ | E-FID↓ | FID↓ | FVD↓ | Sync-C↑ | Sync-D↓ | SSIM↑ | E-FID↓ |
| Audio2Head | 76.08 | 1417.65 | 3.16 | 17.62 | 0.572 | 3.81 | 127.30 | 1882.64 | 1.96 | 17.36 | 0.391 | 8.42 |
| V-Express | 57.14 | 1152.29 | 5.05 | 11.68 | 0.706 | 1.83 | 98.07 | 1465.26 | 3.71 | 13.41 | 0.514 | 5.18 |
| AniPortrait | 54.81 | 1072.63 | 5.40 | 11.39 | 0.727 | 1.95 | 94.25 | 1260.74 | 3.98 | 12.88 | 0.536 | 4.91 |
| SadTalker | 52.77 | 956.24 | 5.73 | 10.65 | 0.736 | 1.87 | 88.22 | 1055.49 | 4.05 | 11.20 | 0.565 | 4.66 |
| Hallo | 37.29 | 616.04 | 6.33 | 8.64 | 0.774 | 1.67 | 72.46 | 907.60 | 6.48 | 8.61 | 0.620 | 2.93 |
| EchoMimic | 31.44 | 595.17 | 6.96 | 8.59 | 0.782 | 1.64 | 71.47 | 893.28 | 6.70 | 8.45 | 0.637 | 2.81 |
| MegActor-$\Sigma$ | 31.37 | 586.10 | 6.87 | 8.55 | 0.778 | 1.62 | 70.82 | 875.21 | 6.77 | 8.32 | 0.634 | 2.74 |
| **MCDM (Ours)** | **26.45** | **543.28** | **7.49** | **8.04** | **0.824** | **1.51** | **67.29** | **784.53** | **7.25** | **7.84** | **0.662** | **2.31** |

*Table 2.* **Quantitative comparisons on HDTF and CelebV-HQ.** MCDM achieves the top results across all metrics, with best in **bold**.

| Method | FID↓ | FVD↓ | Sync-C↑ | Sync-D↓ | SSIM↑ | E-FID↓ |
|---|---|---|---|---|---|---|
| Audio2Head | 87.21 | 1836.25 | 2.32 | 13.92 | 0.613 | 3.12 |
| V-Express | 62.18 | 1324.57 | 5.45 | 9.04 | 0.674 | 2.81 |
| AniPortrait | 56.11 | 954.91 | 6.37 | 8.29 | 0.706 | 2.60 |
| SadTalker | 52.77 | 847.20 | 6.94 | 7.92 | 0.724 | 2.49 |
| Hallo | 51.35 | 792.38 | 6.85 | 7.65 | 0.728 | 2.35 |
| EchoMimic | 49.20 | 751.44 | 7.06 | 7.18 | 0.737 | 2.31 |
| MegActor-$\Sigma$ | 48.57 | 724.40 | 7.22 | 7.14 | 0.745 | 2.29 |
| **MCDM (Ours)** | **42.08** | **656.71** | **7.84** | **6.69** | **0.779** | **1.97** |

*Table 3.* **Quantitative comparisons on TalkingFace-Wild.** MCDM achieves a significant advantage over other methods.

2004) measures structural consistency between ground truth and generated videos, while E-FID (Deng et al., 2019) provides a refined image fidelity evaluation based on Inception network features.

**Implementations.** The experiments are conducted on a computing platform equipped with 8 NVIDIA V100 GPUs. Training is performed in three stages, with each stage consisting of 30,000 iterations and a batch size of 4. Video data is processed at a resolution of $512 \times 512$. The learning rate is fixed at $1 \times 10^{-5}$ across all stages, and the AdamW optimizer is employed to stabilize training. Each training clip comprised 16 video frames. In the archived-clip motion-prior module, we set $\alpha = 16$, $m = 256$, and $n = 16$. In the present-clip motion-prior diffusion model, the number of layers $L$ is set to 8, and the weighting factor $\alpha$ in Eq. 5 is configured to 0.1 to balance the influence of prior motion information. This setup is chosen to optimize long-term identity preservation and enhance motion consistency within generated TalkingFace videos.

### 4.2. Main Results

We compare our method with several SOTA methods, including Audio2Head (Wang et al., 2021a), V-Express (Wang et al., 2024b), AniPortrait (Wei et al., 2024), SadTalker (Zhang et al., 2023), Hallo (Xu et al., 2024), EchoMimic (Chen et al., 2024), and MegActor-$\Sigma$ (Yang et al., 2024), from quantitative, qualitative, and user study. **Unless otherwise specified,** all methods do not use landmarks to ensure a fair comparison.

**Quantitative Evaluation.** Table 2 presents a quantitative

comparison on the HDTF (Zhang et al., 2021) and CelebV-HQ (Zhu et al., 2022), illustrating the overall superior performance of diffusion-based methods compared to GAN-based methods. Our proposed MCDM achieves the best scores across all metrics, outperforming existing diffusion-based approaches. Specifically, MCDM achieves superior lip-sync accuracy, reflected in higher Sync-C and lower Sync-D scores, outperforming methods like EchoMimic (Chen et al., 2024) and MegActor-$\Sigma$ (Yang et al., 2024), which show notable declines in synchronization quality. MCDM's outstanding SSIM and E-FID scores also highlight its ability to generate visually appealing, temporally consistent content with precise lip synchronization.

Table 3 summarizes the quantitative performance on the proposed TalkingFace-Wild dataset. Consistent with results on HDTF (Zhang et al., 2021) and CelebV-HQ (Zhu et al., 2022), MCDM surpasses all competing SOTA methods across evaluation metrics, demonstrating marked improvements in visual quality and temporal consistency. Achieving the best FID, FVD, and an E-FID of 1.97, MCDM shows strong capability in generating high-fidelity TalkingFace videos under diverse conditions, effectively maintaining temporal coherence across audio, expressions, and lip synchronization.

**Qualitative Evaluation.** Figure 3 provides a qualitative comparison of our method against other SOTA approaches. Compared to V-Express (Wang et al., 2024b) and EchoMimic (Chen et al., 2024), our approach shows superior head and lip synchronization, benefiting from the audio-visual consistency introduced by motion priors. Additionally, unlike Hallo (Xu et al., 2024) and MegActor-$\Sigma$ (Yang et al., 2024), Our method accurately captures subtle facial actions, including blinks and expression nuances through the archived-clip, while better preserving identity consistency. Overall, our approach demonstrates the best visual results.

**User Study.** The quantitative and qualitative comparisons underscore the substantial advantages of our proposed MCDM in generating consistent TalkingFace videos. To further evaluate video quality, we conduct a user study, focusing on identity consistency, motion synchronization, and overall video quality. We randomly selected 10 cases, shuf-

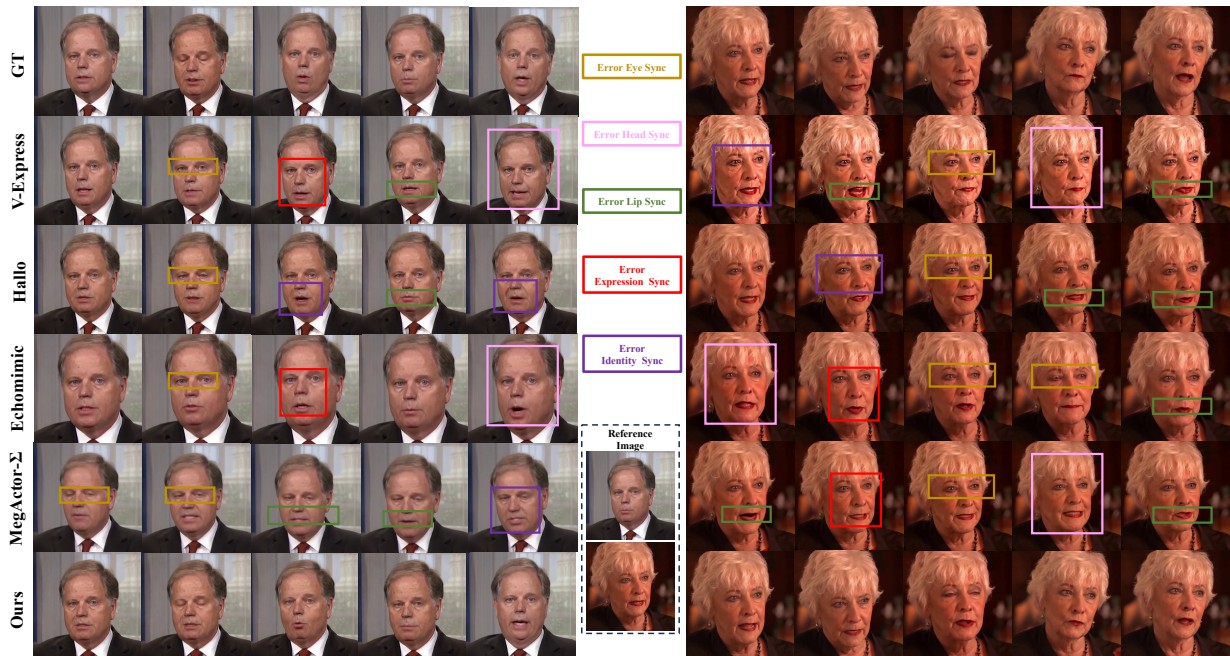

*Figure 3.* **Qualitative comparison on HDTF and CelebV-HQ.** Our method achieves the best generation results, particularly in identity consistency and motion detail.

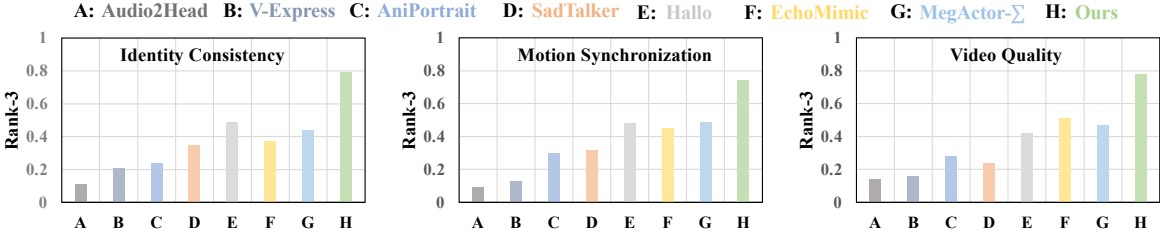

*Figure 4.* **User study results** of identity consistency, motion synchronization, and video quality. Higher values indicate better performance.

| Method | FID↓ | FVD↓ | Sync-C↑ | Sync-D↓ | SSIM↑ | E-FID↓ |
|---|---|---|---|---|---|---|
| w/o $F_a$ | 46.25 | 708.93 | 7.37 | 7.05 | 0.749 | 2.25 |
| w/o $F_{pc}$ | 45.63 | 684.20 | 7.49 | 6.97 | 0.758 | 2.13 |
| w/o $MTA$ | 44.27 | 671.05 | 7.62 | 6.84 | 0.771 | 2.04 |
| **Ours** | **42.08** | **656.71** | **7.84** | **6.69** | **0.779** | **1.97** |

*Table 4.* **Ablation results** on the TalkingFace-Wild dataset.

fled the generated videos from each method, and recruited 20 participants (10 male, 10 female) to provide rank-3 preferences. From Figure 4, our method consistently achieved the highest scores across all metrics in the user preference evaluation. This user study highlights the significant advantage of our approach in user-centric TalkingFace generation.

### 4.3. Ablation Results

We conduct an ablation study to assess the impact of each component in our method. Table 4 shows the results: w/o $F_a$ omits historical frame information, w/o $F_{pc}$ adds an audio attention module for audio feature input, and w/o $MTA$ applies a standard temporal attention module.

**Archived-Clip Motion-Prior.** The results in Table 4 show that removing historical frame information (w/o $F_a$) significantly degrades performance across all metrics, underscoring the importance of the archived-clip motion-prior. To further assess the effect of $F_a$ on long-term generation, we visualized frames 30, 300, 1800, 3600, and 7200 with corresponding SSIM scores, as shown in Figure 5. Figure 5(a) indicates that without the archived-clip (w/o $F_a$), identity consistency worsens with frame progression, resulting in visible artifacts and inconsistencies in head, mouth, and expression. In Figure 5(b), the SSIM scores highlight error accumulation increases with frame count, showing a rapid decline in (w/o $F_a$), while (w/ $F_a$) remains stable at a higher value. These findings validate the effectiveness of the archived-clip motion-prior in preserving both identity and temporal coherence over extended sequences.

**Present-Clip Motion-Prior.** Similarly, excluding the present-clip motion-prior and injecting audio information directly via audio attention (w/o $F_{pc}$) leads to a drop in performance across all metrics. This decline highlights the

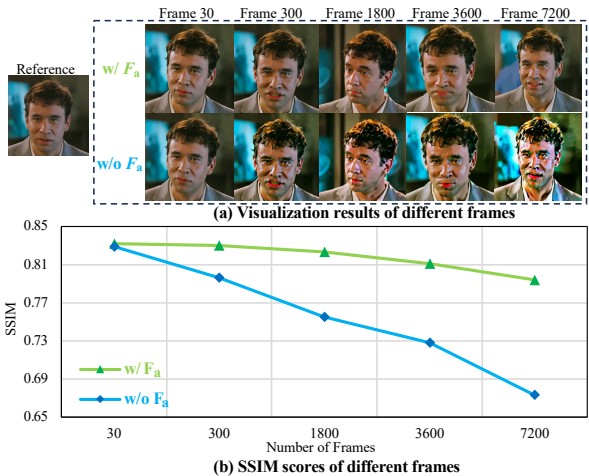

(a) Visualization results of different frames

(b) SSIM scores of different frames

*Figure 5.* **Visualization results and SSIM scores** during long-term generation. We find that w/ $F_a$ offers a distinct advantage in maintaining both identity and contextual consistency.

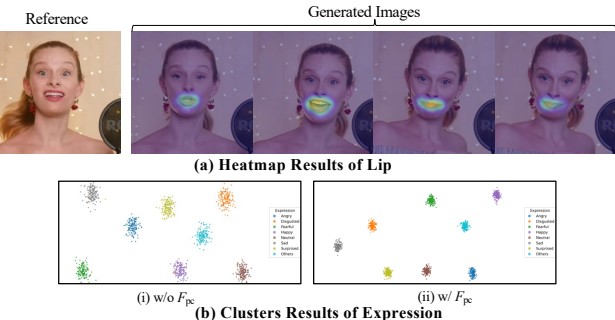

(a) Heatmap Results of Lip

(i) w/o $F_{pc}$      (ii) w/ $F_{pc}$

(b) Clusters Results of Expression

*Figure 6.* **Lip heatmap and expression cluster.** We find that w/ $F_p$ effectively tracks the lip region and conveys expressions.

effectiveness of the present-clip motion-prior in leveraging multimodal causality and temporal interactions to decouple and predict motion states, including expressions, lip movement, and head motion (see Table 4). To further validate this decoupling capability, we visualize heatmaps of the predicted lip tokens, as shown in Figure 6(a), where the present-clip motion-prior accurately localizes and tracks lip motion. For expression decoupling, t-SNE (Van der Maaten & Hinton, 2008) visualization of expression tokens reveals tighter clustering within each of the eight distinct emotion categories when using the present-clip motion-prior, indicating improved separation of emotional content from audio input.

**Memory-Efficient Temporal Attention.** Following the standard approach (Hu, 2024), we replace the proposed memory-efficient temporal attention with conventional temporal attention by directly summing $F_{ac}^{ref}$ and $F_{pc}^{ref}$. As shown in Table 4, this modification significantly degrades performance across all metrics. This drop in quality is primarily due to the absence of an update mechanism, which introduces gaps between the archived clip and the present clip, compromising video smoothness. Next, we analyzed

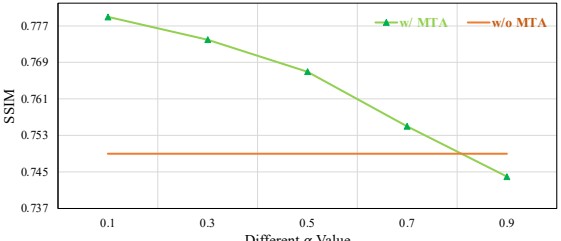

*Figure 7.* **SSIM results for different $\alpha$ values.** Our method performs comparably well when the $\alpha$ value is smaller than 0.9.

| Method | FID↓ | FVD↓ | Sync-C↑ | Sync-D↓ | SSIM↑ | E-FID↓ |
|--------|------|------|---------|---------|-------|--------|
| B1 | 42.49 | 668.24 | 7.69 | 6.78 | 0.771 | 2.02 |
| B2 | 47.12 | 721.17 | 7.30 | 6.84 | 0.732 | 2.29 |
| **Ours** | **42.08** | **656.71** | **7.84** | **6.69** | **0.779** | **1.97** |

*Table 5.* **More results** of variant MCDM.

the effect of different $\alpha$ values in Eq. 5, which control the update rate, on the model's SSIM performance, as shown in Figure 7. We observed that as $\alpha$ increases, SSIM gradually declines. When $\alpha$ is below 0.9, our approach significantly outperforms the w/o $WTA$ configuration. However, at $\alpha = 0.9$, the performance is weaker than w/o $WTA$, due to the excessive accumulation of historical frame information and a reduced proportion of the present clip. Consequently, we set $\alpha = 0.1$ as the default value in this paper.

**More Results.** Table 5 evaluates different design variants. In B1, Q-Former (Li et al., 2023) replaces frame-aligned attention, while in B2, Reference UNet (Hu, 2024) substitutes VAE with Reference UNet, omitting archived-clip information. Results show that frame-aligned attention outperforms Q-Former by effectively capturing temporal context and integrating long-term dependencies. Additionally, using a frozen VAE with a trainable patchify layer proves to be an efficient alternative to the conventional Reference UNet.

## 5. Conclusion

We presented the **M**otion-priors **C**onditional **D**iffusion **M**odel (**MCDM**) to address the challenges of long-term TalkingFace generation by achieving robust identity consistency and motion continuity. MCDM integrates three key innovations: an archived-clip motion-prior to enhance identity representation, a present-clip motion-prior diffusion model for accurate motion prediction, and a memory-efficient temporal attention to mitigate error accumulation over extended sequences. Additionally, we introduced the TalkingFace-Wild dataset, offering over 200 hours of multilingual video data across diverse scenarios. Experimental results demonstrate the effectiveness of MCDM, setting new benchmarks in long-term TalkingFace generation.

# Acknowledgements

This work is supported by the Major Research Program of Jiangsu Province (Grant No. BG2024042). We would also like to thank *Silicon Intelligence*[4] for their support and collaboration.

# Impact Statement

This paper presents the MCDM model, designed to enhance identity and temporal consistency in long-term TalkingFace generation. While MCDM contributes to the advancement of generative modeling, we recognize the potential ethical concerns, including the risks of misuse for creating deceptive content or spreading misinformation. We emphasize the importance of transparency in AI development and support the integration of detection frameworks to mitigate these risks. In alignment with ongoing efforts in responsible AI, we aim to ensure that the benefits of our work are balanced with its ethical implications, promoting safe and constructive applications in society.

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

## Supplementary Material

This supplementary material offers a more detailed exploration of the experiments and methodologies presented in the main paper. Section A introduces a comprehensive set of symbols and definitions to enhance understanding. Section B delves into the implementation details of our proposed method. Section C provides additional experimental results. Sections D and E respectively discuss comparisons with concurrent works and future work.

## A. Some Notations and Definitions

The notations and definitions used throughout this paper are summarized in Table 6.

*Table 6.* Main notations and definitions used in this paper.

| Notation | Definition |
| --- | --- |
| $z_0$ | Target image |
| $t$ | Timestep |
| $z_t$ | Noisy data at step $t$ |
| $c$ | Pose or audio condition |
| $\epsilon$ | Gaussian noise |
| $\epsilon_\theta$ | Diffusion model |
| $X_{ref}$ | Reference frame |
| $X_{arch}$ | Archived clip |
| $F_x$ | Feature of the reference frame |
| $F_a$ | Feature of the archived clip |
| $F_{ac}$ | Feature of archived-clip motion prior |
| $F_p$ | Feature of present-clip motion prior |
| $F_{ac}^{ref}$ | Feature of archived-clip reference attention |
| $F_{pc}^{ref}$ | Feature of present-clip reference attention |

## B. More Details

### B.1. Archived-Clip Motion-Prior Module

**VAE Encoder.** Existing TalkingFace methods (Wang et al., 2024b; Xu et al., 2024; Chen et al., 2024) often rely on Reference UNet (Hu, 2024) to inject and learn identity features from a reference frame. However, training a standalone Reference UNet demands substantial parameters and incurs significant computational costs, restricting its applicability to efficient generation tasks. In contrast, our approach employs a frozen VAE to encode reference images, streamlining the generation process via latent feature projection. By leveraging the pretrained encoding capabilities of the VAE, our method not only preserves identity consistency but also significantly enhances computational efficiency. As demonstrated in Tables 2 and 3 of the main manuscript, the proposed VAE-based approach outperforms traditional Reference UNet methods in identity preservation. Moreover, Reference UNet exhibits considerable limitations in integrating archived frames, rendering it less effective for long-sequence generation. Our archived-clip motion-prior module seamlessly incorporates archived frames, enabling high-quality long-term TalkingFace generation. Results in Table 7 of the main manuscript further highlight the robustness and effectiveness of our method.

**Frame-Aligned Attention.** We propose frame-aligned attention to align the reference frame with the archived frames to incorporate additional archived frames. The frame-aligned attention overcomes the limitations of traditional Q-Former (Li et al., 2023) in preserving temporal consistency and identity integrity. Unlike Q-former, which is primarily tailored for short-sequence multimodal alignment, frame-aligned attention dynamically aligns feature tokens between reference and archived frames, enabling precise modeling of static identity features and dynamic temporal dependencies. This method achieves superior performance in TalkingFace generation, as evidenced in Table 7 of the main manuscript. Additionally, frame-aligned attention employs frame-wise attention to optimize computational efficiency, making it highly effective for long-sequence tasks. In contrast, Q-Former relies on global attention, concatenating features from all frames, leading to computational costs that scale linearly with sequence length and insufficient temporal modeling. The frame-aligned attention

From the following videos, please select the top 3 in each category: **Identity Consistency**, **Motion Synchronization**, and **Video Quality**.

**Identity Consistency**: Look for how well the person's facial features, skin tone, and overall appearance stay the same throughout the video, without any noticeable changes.

**Motion Synchronization**: Focus on whether the lip movements, head movements, and facial expressions match the audio and seem natural over time.

**Video Quality**: Check the clarity, detail, and smoothness of the video. Look for sharp resolution, realistic textures, and smooth transitions between frames.

7. Please select the top 3 from the following:

A.mp4
120.09 KB

B.mp4
130.22 KB

C.mp4
198.6 KB

D.mp4
163.51 KB

E.mp4
86.14 KB

F.mp4
70.34 KB

G.mp4
133.72 KB

H.mp4
146.47 KB

**Identity Consistency** :

**Motion Synchronization**:

**Video Quality** :

*Figure 8.* **Results of different expression.** Our proposed MCDM demonstrates remarkably realistic performance across four common emotional audio categories.

seamlessly integrates archived frames, significantly improving both temporal consistency and identity preservation.

### B.2. Present-Clip Motion-Prior Diffusion Model

**Lip and Express Encoders.** Lip motion is driven by speech content, while emotions influence expression motion. The entanglement of these components often results in inconsistencies, such as a smiling expression paired with tense lip movements, undermining the realism and naturalness of the generated results. We propose the present-clip motion-prior diffusion model, which disentangles lip and expression motion features from lip and express encoders and predicts them independently. The training process involves constructing a pseudo-labeled dataset of paired videos with consistent expressions but varying lip motions, or vice versa, to facilitate disentangled learning. A 4-layer Transformer-based lip encoder and express encoder are used to extract speech-driven lip motion and emotion-driven expression features separately. These features are integrated into a motion decoder to reconstruct facial motion, including landmarks and local deformations. The training is optimized with a total loss function comprising self-reconstruction, cross-reconstruction, and consistency losses, ensuring effective disentanglement and accurate motion reconstruction.

**Temporal Interaction Transformer Block.** In text-to-image (T2I) generation, Kandinsky (Razzhigaev et al., 2023) has demonstrated the effectiveness of predicting image features to improve model quality. However, maintaining temporal consistency is crucial for TalkingFace generation, where prior T2I models primarily focus on predicting features for static conditions (e.g., image) and lack the capability to model motion continuity across sequences. To address this limitation, we propose the present-clip motion-prior diffusion model, which presents a temporal interaction transformer block to ensure

| Method | 30 | 300 | 1800 | 3600 | 7200 |
|---|---|---|---|---|---|
| V-Express (Wang et al., 2024b) | 1.43 | 15.7 | N/A | N/A | N/A |
| Vanilla (Ours) | 0.97 | 8.6 | 54.1 | 122.5 | 264.6 |
| **MCDM (Ours)** | **0.75** | **5.7** | **27.3** | **46.9** | **74.5** |

*Table 7.* **Results of inference speed.** As the number of frames increases, the advantage of our method becomes more pronounced. "N/A" denotes that the GPU encountered an overflow issue and was unable to complete the task. The unit is minutes.

temporal coherence. Specifically, multimodal correlation tokens and noise-injected decoupled motion features are extracted using corresponding encoders and a feature-wise linear modulation (FiLM) layer (Perez et al., 2018). These tokens are concatenated along the token dimension and represented as a 4D tensor $x \in \mathbb{R}^{b \times f \times n \times d}$, where $b$, $f$, $n$, and $d$ denote batch size, temporal length, token count, and token dimension, respectively. During processing, the temporal dimension $f$ is reshaped into the batch size $b$ within a multimodal causal transformer block (Peebles & Xie, 2023), allowing independent frame-wise processing. After passing through the multimodal causal transformer block, the features are reshaped back into a 4D tensor to preserve temporal relationships. The temporal interaction transformer further reshapes the token dimension $n$ into the batch size $b$ to learn and maintain temporal consistency for TalkingFace generation before reshaping it back, ignoring the temporal dimension. The temporal interaction transformer block employs multiple self-attention modules to guide self-attention along the temporal dimension $f$, effectively capturing motion features.

### B.3. Memory-Efficient Temporal Attention

**Fast Attention.** In memory-efficient temporal attention, both self-attention (Guo et al., 2023) and fast attention (Choromanski et al., 2020) are viable options. However, fast attention is particularly suited for capturing long-term temporal dependencies. As sequence length increases, traditional mechanisms like cross-attention face scalability challenges, resulting in higher computational and memory costs. Fast attention, with its optimized computation, allows efficient modeling of extended temporal relationships, making it ideal for tasks requiring consistent and coherent motion across a large number of frames while maintaining computational feasibility and temporal fidelity.

### B.4. User Study

As shown in Figure 8, we conducted a user study to comprehensively evaluate the quality of the generated videos, focusing on three key metrics: identity consistency, motion synchronization, and video quality. The detailed procedure is as follows:

**Data Preparation.** We randomly selected 10 samples from the test set, where each sample includes one input and its corresponding target video. For each sample, videos were generated using eight different methods, resulting in a total of 80 video clips (10 samples × 8 methods). These videos were randomly shuffled to ensure anonymity and fairness during the evaluation.

**Evaluation Rules.** We recruited 20 participants, comprising 10 males and 10 females, from diverse backgrounds with adequate visual perception abilities. Before the evaluation, participants were given detailed instructions on the evaluation criteria:

- **Identity Consistency**: Assess whether the facial identity remains consistent across frames.

- **Motion Synchronization**: Evaluate the alignment between lip motion and audio input.

- **Video Quality**: Consider the naturalness, smoothness, and realism of the video.

Participants were asked to rank the videos using the **rank-3** rule, selecting the top three videos from each set and ordering them by preference (1 being the best, 3 being the third best). To ensure accurate evaluations, participants were allowed to rewatch videos as needed. *The study followed ethical principles outlined in relevant guidelines. All participants provided informed consent prior to the study, and their feedback was anonymized to protect privacy.*

## C. More Results

**Comparisons in Inference Speed.** To further evaluate inference speed, we conducted an additional experiment, as shown in Table 7. Using the representative Reference-based architecture V-Express (Wang et al., 2024b) as a baseline, we kept all other

| Settings | FID↓ | FVD↓ | Sync-C↑ | Sync-D↓ | SSIM↑ | E-FID↓ |
|---|---|---|---|---|---|---|
| 2 | 45.72 | 704.26 | 7.39 | 7.03 | 0.754 | 2.19 |
| 4 | 44.53 | 694.61 | 7.55 | 6.93 | 0.767 | 2.08 |
| 8 | 43.68 | 678.49 | 7.68 | 6.81 | 0.773 | 2.03 |
| 16 | **42.08** | **656.71** | **7.84** | **6.69** | **0.779** | **1.97** |

*Table 8.* **Results of different archived frames.** We observed that increasing the number of archived frames significantly improves the quality of the generated results.

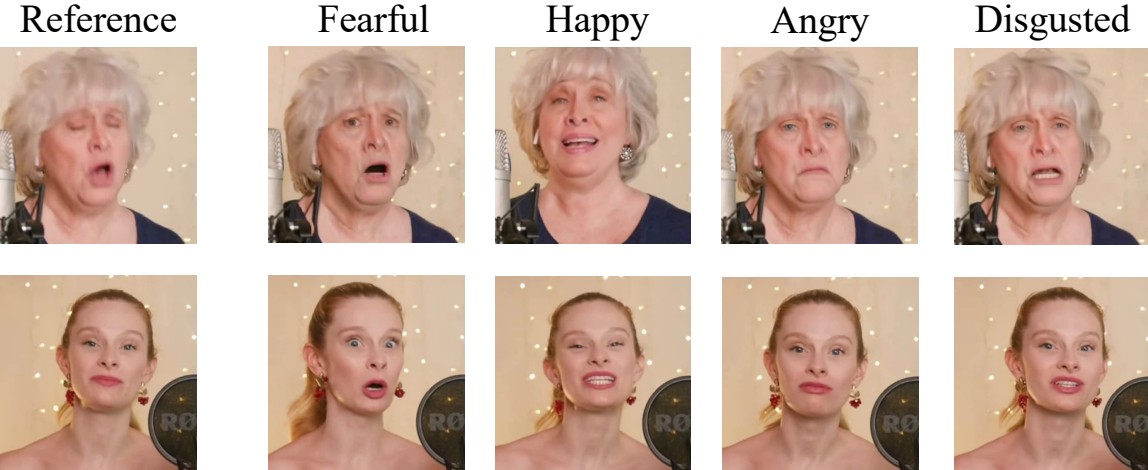

Reference      Fearful      Happy      Angry      Disgusted

*Figure 9.* **Results of different expression.** Our proposed MCDM demonstrates remarkably realistic performance across four common emotional audio categories.

settings of MCDM unchanged and replaced fast attention (Choromanski et al., 2020) with the commonly used self-attention (Vanilla) (Guo et al., 2023) and our proposed MCDM. We measured the time required to generate 30, 300, 1800, 3600, and 7200 frames (in minutes). The results show that for 30 and 300 frames, V-Express (Wang et al., 2024b) significantly lags behind both Vanilla (Ours) and MCDM (Ours). Notably, as the number of frames increases, the Reference-based V-Express encounters GPU memory overflow, rendering it unable to handle more extensive sequences. In contrast, our proposed MCDM demonstrates increasing efficiency advantages over Vanilla as the frame count grows. Specifically, for 7200 frames, our method achieves nearly 4× speedup compared to Vanilla.

**Influence of Different Archived Frame Numbers.** To investigate the influence of archived clip length, we conducted experiments using 2, 4, 8, and 16 frames while keeping all other settings fixed. As presented in Table 8, increasing the number of archived frames leads to notable improvements in both identity consistency and temporal coherence. This result highlights the critical role of archived frames and further demonstrates the effectiveness of our proposed archived-clip motion-prior module.

**Results of Different Emotional Audio.** To comprehensively evaluate the capability of our proposed MCDM in audio disentanglement tasks, we select four classic emotional audio categories as inputs: 'fearful,' 'happy,' 'angry,' and 'disgusted.' These audio clips drive a single reference image to generate the corresponding facial dynamics, as illustrated in Figure 9. The results demonstrate that MCDM achieves remarkably lifelike performance across all four emotional audio types, with outstanding detail fidelity. Subtle variations in eye expressions and mouth movements further highlight the model's ability to capture intricate facial features and emotional nuances.

**Comparisons with T2I Prior Diffusion Model.** To validate the effectiveness of the proposed present-clip motion-prior diffusion model, we compare its performance with the Kandinsky (Razzhigaev et al., 2023). Kandinsky's results are obtained by independently predicting features and then averaging them. As shown in Table 9, the present-clip motion-prior diffusion model significantly outperforms Kandinsky in generating prior features. This improvement is attributed to the model's ability to capture temporal consistency, whereas Kandinsky only predicts static image features. This inherent conflict with the temporal consistency required for TalkingFace generation limits Kandinsky's effectiveness in modeling motion sequences.

**Results of Long-Term TalkingFace Generation.** Figure 10 illustrates the long-term TalkingFace generation results of the proposed MCDM. The generated sequences demonstrate consistent identity preservation throughout the extended duration.

| Methods | Cosine Similarity↑ |
|---|---|
| Kandinsky | 0.656 |
| **MCDM (Ours)** | **0.947** |

*Table 9.* **Results of prior model.** We compute the average cosine similarity of motion features, revealing that MCDM demonstrates a significant advantage in predicting motion features.

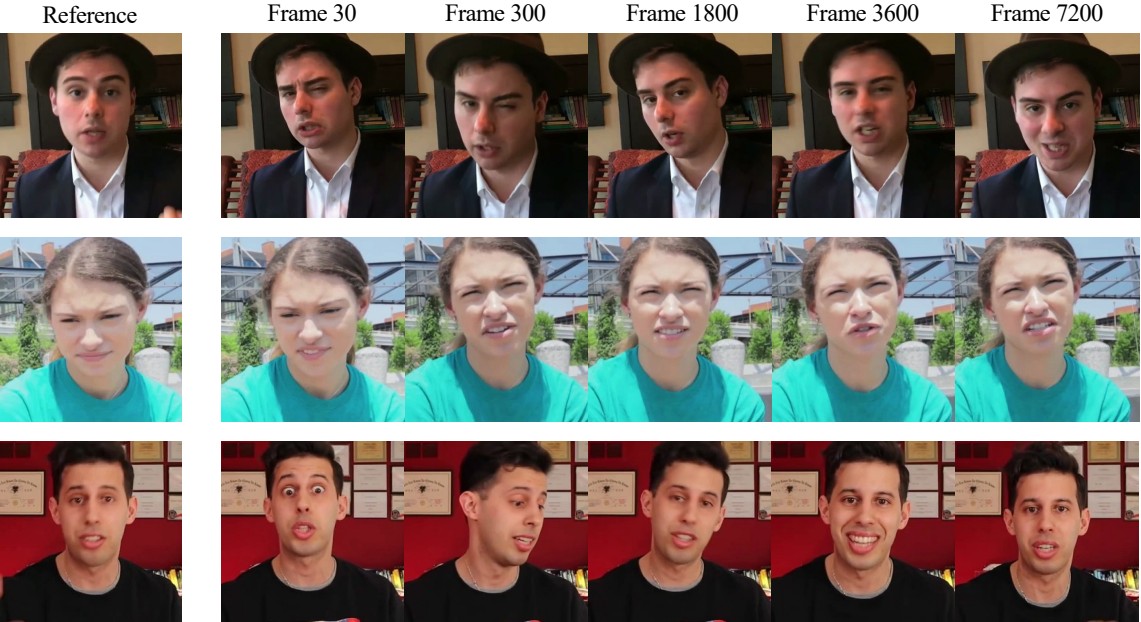

*Figure 10.* **Results of long-term TalkingFace generation.** MCDM maintains long-term TalkingFace consistency and enables natural head pose transitions.

Additionally, the transitions in head poses appear smooth and natural, further validating the effectiveness of MCDM.

## D. Discussion of Concurrent Work

Alongside our work, there are two notable concurrent studies, Loopy (Jiang et al., 2024) and Hallo2 (Cui et al., 2024) for TalkingFace generation. Since both are concurrent works without available code, we limit our discussion to their methodologies. Loopy (Jiang et al., 2024) introduces inter- and intra-clip temporal modules to capture long-term motion dependencies, focusing on natural and unconstrained motion without spatial constraints. Hallo2 (Cui et al., 2024) extends portrait animation to long-duration videos by employing patch-drop and Gaussian noise augmentations to address temporal artifacts and improve consistency. In contrast, our proposed MCDM introduces the Archived-Clip Motion-Prior and Present-Clip Motion-Prior mechanisms, explicitly disentangling identity and motion features while leveraging memory-efficient temporal attention. Unlike Loopy's emphasis on naturalness or Hallo2's augmentation-based strategies, MCDM achieves superior identity preservation and temporal coherence, particularly in long-frame scenarios, while maintaining computational efficiency.

## E. Future Work

In our experiments, the proposed motion-prior conditional diffusion model (MCDM) effectively demonstrates that combining archived and present motion priors significantly improves identity consistency and temporal coherence in long-sequence TalkingFace generation tasks. This enhancement leads to a notable improvement in the overall quality of the generated videos. MCDM is the first approach to explicitly focus on motion priors while eliminating dependency on Reference UNet, thereby addressing GPU memory constraints. Additionally, its simple and modular design ensures ease of reproduction. Nonetheless, several directions for future exploration remain. These include extending the method to tasks such as "animate anyone" for dance generation, full-body digital human synthesis, and, in particular, scenarios involving multiple reference images. In these contexts, the continued reliance on Reference UNet presents a limitation, marking an important area for future research.

