# OpenReview forum: "Long-Term TalkingFace Generation via Motion-Prior Conditional Diffusion Model"
_ICML.cc/2025/Conference — ICML 2025 poster_

### Official Review · Reviewer_kTry · 2025-03-11

**Overall Recommendation:** 3

**Summary:**

This paper looks at generating long-form videos of talking faces with a view to improving the overall expressivity and naturalness of the head motion / facial expression accompanying the speech. A diffusion model is used for generation that is conditioned on various cues, which include historical motion context from other videos, local motion context for the current sequence, an encoding of the lips and facial expression, and optionally the facial landmarks to help guide the motion generation.

**Claims And Evidence:**

Some aspects of the expressiveness is problematic in the example video.  For example, the head motion in the long-form sequence when the speaker rotates their head to their left and then back to the camera looks slow and seemingly unnatural.  Also, the eyes are staring and look unnatural.  The speech articulation looks very good, but I find other aspects of the selected videos do not support the claim that the work is significantly more expressive.

**Essential References Not Discussed:**

Modern literature is cited, but I am not sure it is fair to call the cited examples in the introduction “early methods”.  You can go back to the late 1990s to find work on creating expressive talking faces (albeit not deep learning / neural based methods).  The paper does cite the relevant recent literature.

**Experimental Designs Or Analyses:**

I have questions about the subjective evaluation.   Specifically:  1) are viewers expected to watch all eight sequences and select the top three?  This is a placing a high load on the viewers to keep track of which is the best three, and going back and forth to compare.  2) How long are viewers given to complete the task?   2) The text mentions randomization — can I confirm that the video labeled as A.mp4 is not always the same system that generates the A sequences?  The text seems to suggest that the sequences are randomized, but the filename shown (A.mp4) should not be the same for a given system each time as this risks introducing bias.

**Methods And Evaluation Criteria:**

The authors curate a new dataset for benchmarking, which they will release.  Other standard datasets have been used.  The approach is tested against a number of state of the art methods.  Also, standard metrics are used to measure performance.

**Other Comments Or Suggestions:**

Throughout the paper you use TalkingFace as one word — you should use “talking face” when discussing talking faces in general.

In Figure 1 for the present-clip motion-prior diffusion model:  you refer to the “express encoder”, which should read “expression encoder”.

Line 132: “a archived clip” > “an archived clip”.
Line 151:  “limited history frame” > “limited frame history”.
Line 182: “and express encoder” > “and expression encoder”.
Line 196: “Lip and Express Encoders” > “Lip and Expression Encoders”.
Line 198: “lip and express encoders” > “lip and expression encoders”.
Line 316: remove the underline.

Appendix A is useful.

**Other Strengths And Weaknesses:**

Strengths:
+ The objective metrics suggest the approach compares favorably against all baseline on all three datasets.
+ The code and data will be released.

Weaknesses:
- There are artifacts in the output from the proposed system, especially around the chin.  I observed this in two of the examples.
- See the questions regarding the subjective evaluation.  Why not do something more typical, like MOS?

**Questions For Authors:**

Q1:  The paper states that conditioning on too many past frames for the generated sequence is computationally expensive.  Why is it more expensive than conditioning on sequences from the archive?

Q2:  If the landmark tokens are dropped, are they simply replaced with zeros?

Q3:  On line 299: for the archived-clip motion-prior module, the text states \alpha=16.  There is no \alpha mentioned in the description of this module in Section 3.2.  Rather this appears in the description of the memory-efficient temporal attention in Section 3.4.

Q4:  For the example showing the importance of Fa the text reads:  “Figure 5(a) indicates that without the archived-clip (w/o Fa), identity consistency worsens with frame progression“ — this does not look like a shift in identity, but rather image artifacts being introduced.

**Relation To Broader Scientific Literature:**

Using the library of archived motion sequences to condition the generator is novel.  Similar work has been done encoding the expressiveness and eyes separately, but the application here is different from what I have seen.

**Theoretical Claims:**

There are no theoretical claims in the paper.

---

> ### Author Rebuttal · Authors · 2025-04-01
>
> Dear Reviewer kTry:
>
> **Thank you very much for  the detailed comments. We address your concerns as follows.**
>
> **Q1: Lack of Clarity in Technical Descriptions**
>
> **Response:** Thank you for the helpful comments. We address each point below:
>
> 1. **Memory Cost of Past Frames vs. Archived Sequences:**
>    Standard temporal self-attention stores all key-value pairs, resulting in quadratic memory growth with the number of frames. In contrast, our Archived-Clip Motion-Prior Module integrates with Memory-Efficient Temporal Attention (MTA), which maintains a fixed-size memory via incremental updates, avoiding GPU memory overflow in long-term generation. Please see our experimentalanalysis in Q2 of R2 (Reviwer S3hq).
> 2. **Landmark Token Drop:**
>    Yes, we replace dropped landmark tokens with zeros.
> 3. **Typo in Line 299:**
>    Sorry, the symbol “\alpha” was a typo, it should be the variable name “a” and is unrelated to Section 3.4.
> 4. **Interpretation of Figure 5(a):**
>    We agree with your observation. Our intent was not to suggest identity drift. As stated in line 375, the absence of archived-clip priors leads to visible artifacts and inconsistencies in head, mouth, and expression, not identity change. We will clarify this wording in the revised version.
>
> **Q2: Issues in Subjective Evaluation Design**
>
> **Response:** Thank you for your thoughtful comments regarding the user study design. We address each concern below:
>
> 1. **Rank Selection Protocol:**
>    Following prior works [1,2], we adopted a ranking-based evaluation strategy to effectively capture relative user preferences. While this approach may introduce some cognitive load, participants were allowed unlimited time and could freely replay videos to ensure fair and consistent judgments.
>
>    [1] Ma et al., "Follow Your Emoji: Fine-Controllable and Expressive Freestyle Portrait Animation", SIGGRAPH Asia 2024.
>
>    [2] Gao et al., "FaceShot: Bring Any Character into Life", ICLR 2025.
> 2. **Time Control:**
>    There was no strict time limit. Participants were allowed to complete the task at their own pace.
> 3. **Randomization and File Naming:**
>    We confirm that the method-to-letter mapping (e.g., A.mp4, B.mp4) was independently randomized for each session. The correspondence was stored using a dictionary to avoid fixed associations and mitigate bias.
> 4. **Addition of MOS Study:**
>    Thank you for your  thoughtful advice.  Following your suggestions, we have additionally conducted a Mean Opinion Score (MOS) study with 20 participants. Each participant rated video quality on a 1–5 scale (5 = Excellent, 1 = Bad). The results are summarized below:
>
> | **Method**       | **MOS Score ↑** |
> |------------------  |------------------|
> | Audio2Head       | 1.6              |
> | V-Express        | 2.6              |
> | AniPortrait      | 2.9              |
> | SadTalker        | 3.2              |
> | Hallo            | 3.8              |
> | EchoMimic        | 3.7              |
> | MegActor-∑       | 3.5              |
> | **Ours (MCDM)**  | **4.3**          |
>
> The experimental results demonstrate that our method achieved the highest MOS score, demonstrating strong user preference in terms of identity consistency and visual quality.
>
> **Q3: Writing and Terminology Issues**
>
> **Response:** Thank you for the reminder. We have fixed all mentioned issues, including terminology corrections, grammar errors, and formatting. We also reviewed the entire paper to correct other typos, table labels, and figure captions for consistency and clarity.
>
> **Q4: Concerns on Expressiveness and Visual Artifacts**
>
> **Response:** Thank you for the valuable feedback. We acknowledge that minor artifacts (e.g., around the chin or eyes) may appear in difficult cases involving large head movements, this remains an open challenge in the community. **Our work takes a meaningful step forward by proposing a memory-based generation framework and releasing a 200-hour multilingual dataset to help mitigate these issues.** Both quantitative and qualitative results validate the motivation of our approach, showing that our method consistently outperforms prior baselines in expressiveness and coherence. **More visual comparisons, including EMO (closed-source, official demo inputs), are provided in [link 1](https://files.catbox.moe/77kqif.mov) and [link 2](https://files.catbox.moe/1p4qo4.mov), demonstrating superior identity consistency and visual quality.**
>
> **Q5: Literature Framing and Terminology**
>
> **Response:** Thank you for the suggestion. We agree that referring to recent deep learning methods as “early” may be misleading. We have revised the introduction to more clearly distinguish early non-neural approaches from modern neural-based methods.
>
> **Lastly, thank you so much for helping us improve the paper and  appreciate your open discussions! Please let us know if you have any  further questions. We are actively available until the end of this  rebuttal period. Looking forward to hearing back from you!**

---

> > ### Comment · Reviewer_kTry · 2025-04-03
> >
> > Thank you for the follow up and the pointers to the rank-based subjective assessment.  Your clarification here about allowing unrestricted viewing and unrestricted time makes sense. I appreciated the additional subjective testing that you ran. I will raise my score.

---

> > > ### Author Response · Authors · 2025-04-03
> > >
> > > Dear Reviewer kTry:
> > >
> > > Thank you sincerely for taking the time to review our rebuttal and for thoughtfully considering our clarifications. We are especially grateful that you found the additional subjective evaluation and explanations helpful. **We truly appreciate your updated score and your constructive feedback throughout the review process,** it has been invaluable in helping us strengthen the quality and clarity of our work.
> > >
> > > **If you have any additional questions or anything you would like to discuss in more detail, please feel free to let us know. We would be more than happy to discuss further and respond promptly.**

---

### Official Review · Reviewer_S3hq · 2025-03-13

**Overall Recommendation:** 3

**Summary:**

In this paper proposed the Motion-Priors Conditional Diffusion Model (MCDM), a diffusion-based framework designed for improved long-term motion generation to resolve the key challenges in existing methods, including lack of identity consistency, limited expression diversity, audio-motion synchronization issues, and error accumulation in long-term facial motion generation.

**Claims And Evidence:**

In this paper claim that the weak correlation between audio and motion complicates the decoupling of identity and motion cues.

However, audio and lip motion are strongly correlated. The author did not clearly whether “motion” refers to lip motion, head motion, or both, which creates ambiguity in evaluating the claim.

**Essential References Not Discussed:**

.

**Experimental Designs Or Analyses:**

The authors introduce Memory-Efficient Temporal Attention (MTA) to improve long-term stability.
However, the authors does not provide the evidences that MTA is more memory-efficient than existing approaches.

No parameter comparison: The paper does not compare the parameter count of MTA with previous temporal attention mechanisms.
No memory usage evaluation: There is no quantitative measurement of GPU memory usage or computational cost (e.g., FLOPs, VRAM consumption).
No baseline comparison: The paper does not include comparisons with existing self-attention methods to demonstrate MTA’s efficiency.

**Methods And Evaluation Criteria:**

The proposed method is well aligned with the problem of speaking face generation.
The evaluation criteria (dataset, metrics) are adequate and widely used in previous work.
However, the lack of efficiency evaluation weakens the memory-efficiency claim.
The lack of direct comparison with reference video-based methods makes it difficult to evaluate the benefits of the archived clip motion first.

Memory-efficient attention lacks complexity analysis to justify the efficiency claim.
The concept of "multi-causality" is mentioned, but there is no explanation of how it exists.
Even experimentally, there was no quantitative analysis of how much improvement was achieved when causality was broken.

**Other Comments Or Suggestions:**

The explanation of “Multimodal Causality” in Section 3.2 is somewhat vague. It would help to provide a more concrete formulation or a visual representation to clarify how causality is modeled between audio and facial motion.

The authors claim that the Memory-Efficient Temporal Attention module is to be more efficient, but the paper does not provide direct comparisons with standard self-attention in terms of computational cost or memory usage. Including FLOPs or VRAM usage in Table 7 would strengthen the claim.

**Other Strengths And Weaknesses:**

* Strengths:

The authors proposed named TalkingFace-Wild, a multilingual dataset with over 200 hours of high-quality video, is a valuable contribution to the research community.

The proposed Archived-Clip Motion-Prior Module effectively integrates long-term motion context, which is a promising direction for improving motion coherence.

* Weaknesses:

Memory efficiency is not empirically validated: No direct comparisons of parameter count, computational complexity (FLOPs), or VRAM consumption are provided.

The user study is limited in number of subjects (20 subjects), which raises concerns about the statistical reliability of the subjective evaluations.

The paper does not compare its approach to reference video-based motion modeling methods, which are commonly used to improve long-term consistency.

**Questions For Authors:**

Memory-Efficient Temporal Attention:

-	The authors mentioned Memory-Efficient Temporal Attention as a motivation, pointing out the GPU memory limitation of existing methods, but no quantitative evaluation was done. Can you provide a table comparing the efficiency of this method with existing methods?

User Study:

-	The paper conducts a user study with 20 subjects, which may raise concerns about the statistical reliability of the results.

**Relation To Broader Scientific Literature:**

This paper tried to address several issues that arise in existing Talking Face Generation.

To address motion distortion and flickering that occur during long-term generation in GAN and diffusion models, we propose the Motion-priors Conditional Diffusion Model (MCDM).

However, existing papers also try to preserve temporal information by using seq2seq specialized models to preserve temporal information well. Is there a reason why this discussion is missing?

**Theoretical Claims:**

Memory-efficient attention lacks complexity analysis to justify the efficiency claim.

The concept of "multi-causality" is mentioned, but there is no explanation of how it exists.
Even experimentally, there was no quantitative analysis of how much improvement was achieved when causality was broken.

---

> ### Author Rebuttal · Authors · 2025-04-01
>
> Dear Reviewer S3hq:
>
> Thank you for your detailed and constructive review. **We appreciate your recognition of our contributions, including the TalkingFace-Wild dataset and the proposed motion-prior framework.**  We address your concerns as follows.
>
> **Q1: Ambiguity in Audio-Motion Correlation**
>
> **Response:** Thank you for pointing this out. The original sentence "*Reliance on static audio features and weak correlations between audio and 'motion' complicate the decoupling of identity and motion cues...*".  'motion' refers specifically to **head motion**, not lip motion. Following [1-2], audio and head motion are only weakly correlated. We have clarified this distinction in the revised text.
>
> [1] Wang et al., "V-express: Conditional dropout for progressive training of portrait video generation"
>
> [2] Chen et al., "Audio-driven Talking-Head Generation with Rhythmic Head Motion" .
>
> **Q2: Missing Complexity and Efficiency Analysis for MTA**
>
> **Response:** Thank you for the detailed feedback. Standard temporal self-attention requires storing all key-value pairs, leading to quadratic memory growth and potential OOM issues in long sequences (e.g., V-Express in Table 7 is marked N/A due to this).
>
> In contrast, our Memory-Efficient Temporal Attention (MTA) maintains a fixed-size memory via incremental updates, enabling stable and efficient long-term modeling.
>
> Furthermore, we provide a comparison (T=4, batch size=1):
>
> | **Method**        | **FLOPs (G) ↓** | **Training GPU Memory (GB) ↓** | **FID ↓** |
> |------------------|------------------|-------------------------------|------------|
> | Self-Attention     | 6.7              | 28.7                          | 46.29      |
> | **MTA (Ours)**     | **3.1**          | **16.4**                      | **42.08**  |
>
> MTA reduces FLOPs by **54%** and GPU memory by **43%**, while also improving FID. We will include this result in the final version.
>
> **Q3: Lack of Explanation for Multimodal Causality**
>
> **Response:** Thank you for highlighting this point. We apologize for the lack of clarity. By *multimodal causality (MC)*, we refer to modeling head motion, lip motion, and expression tokens jointly using a unified transformer conditioned on audio and reference image features. These conditioning tokens are prepended to the sequence and influence all motion tokens uniformly. Unlike modality-specific denoising, our model adopts a shared causal transformer to perform unified denoising across all motion cues. To quantify its effectiveness, we conducted an ablation study:
>
> | **Method**   | **FID ↓** | **FVD ↓** | **Sync-C ↑** | **Sync-D ↓** | **SSIM ↑** | **E-FID ↓** |
> |--------------|-----------|-----------|--------------|--------------|------------|-------------|
> | w/o MC       | 46.03     | 702.18    | 7.42         | 7.03         | 0.754      | 2.19        |
> | **w/ MC**    | **42.08** | **656.71**| **7.84**     | **6.69**     | **0.779**  | **1.97**    |
>
> These results demonstrate that multimodal causality modeling leads to consistent improvements in identity preservation, synchronization, and overall generation quality.
>
> **Q4: Missing Comparison with Reference-Based and Seq2Seq Methods**
>
> **Response:** Thank you for the suggestion. As noted in Q2, reference-based methods (e.g.,EchoMimic) segment long videos to mitigate OOM, but often lead to identity inconsistency between clips. Seq2Seq models (e.g., SadTalker and EMO) lack memory mechanisms and are prone to error accumulation. Our unified memory-based framework avoids these issues by maintaining and updating a fixed-size memory for consistent long-term generation. We have added qualitative comparisons in the [link 1](https://files.catbox.moe/77kqif.mov), and [link 2](https://files.catbox.moe/1p4qo4.mov). which demonstrate superior identity consistency and visual quality.
>
> **Q5: Concerns About the User Study Design**
>
> **Response:** We would like to clarify that in our user study, the mapping between methods and filenames (e.g., A.mp4, B.mp4) was randomly shuffled for each evaluation task, and the correspondence was stored using a dictionary to ensure fair randomization and avoid naming bias.
>
> The initial study involved 20 participants, which aligns with common practice in prior works [3]. To enhance statistical reliability, we extended the study to 40 participants. We made every effort to expand the subject pool within the available timeframe. The results are available at: [link3](https://files.catbox.moe/dm8rio.pdf).
>
> [3] Zhang et al., "SadTalker: Learning Realistic 3D Motion Coefficients for Stylized Audio-Driven Single Image Talking Face Animation".
>
> **In the end, thanks a lot for your detailed comments and thank  you for helping us improve our work! We appreciate your thoughts on our  work and we would be more than happy to discuss more during or after the  rebuttal to explore more in this direction. Please let us know if you  have any further questions. We are actively available until the end of  this rebuttal period.**

---

### Official Review · Reviewer_Vnrq · 2025-03-14

**Overall Recommendation:** 3

**Summary:**

The paper introduces the Motion-priors Conditional Diffusion Model (MCDM) for long-term TalkingFace generation, leveraging archived historical frames and present-clip motion priors through a memory-efficient temporal attention mechanism to achieve robust identity preservation, synchronized expressions, and accurate lip-audio alignment while releasing the multilingual TalkingFace-Wild dataset and demonstrating state-of-the-art performance across benchmarks.

## update after rebuttal
Thanks for your detailed responses. I appreciate the additional experiments, qualitative comparisons, and clarifications, which have improved the quality of the paper. Given these improvements, I am happy to maintain my original rating and support the paper's acceptance.

**Claims And Evidence:**

Yes.

**Essential References Not Discussed:**

No.

**Ethical Review Concerns:**

The research focuses on facial applications, which raises concerns about privacy, security, and human rights, as facial images can be easily modified without individuals' consent.

**Ethics Expertise Needed:**

["Inappropriate Potential Applications & Impact  (e.g., human rights concerns)", "Privacy and Security"]

**Experimental Designs Or Analyses:**

Yes.

**Methods And Evaluation Criteria:**

Yes.

**Other Comments Or Suggestions:**

The writing of the Introduction section needs refinement:

1. The transition and logical flow between the second and third paragraphs, especially the use of logical connectors, should be improved and supported with sufficient evidence. For example,
a) The relationship between GAN-based methods and Diffusion-based models needs to be clarified to ensure coherence.
b) In line 041, the claim that "facial landmarks overly bind poses to the reference image" is unclear — this motion constraint should naturally allow different poses across frames.
c) In line 043, the assertion that overly binding poses limits expression diversity is also questionable. Facial expressions are largely independent of head pose — humans can maintain a fixed pose while still displaying a wide range of expressions.
d) Additionally, the term "static audio feature" is ambiguous — audio features extracted from a time window inherently contain dynamic information.

2. Contributions 2, 3, and 4 should be merged, as they all pertain to technical details and belong to the same category of contributions.

**Other Strengths And Weaknesses:**

Overall, this paper presents an interesting idea and is well-structured, with carefully designed figures. The experiments are rich and comprehensive, and the work meets the standard of ICML. However, I still have several concerns:

1) Ablation Study: The archived motion prior module defaults to using 16 frames of historical information, but the impact of using shorter or longer histories on long-term consistency is not explored. For instance, would 4 frames be sufficient, or would more than 16 frames yield better results? A thorough ablation study on this aspect would strengthen the work.

2) Comparison with Related Work: Although the two related papers [1][2] do not release their code, it would still be valuable to provide qualitative comparisons with them. Since both papers offer demo videos on their official websites, comparisons could be made using their audio and images at least at a qualitative level to better position this work among existing approaches.

3) Concurrent Work: The authors mention that they do not compare with the concurrent method [3] due to the unavailability of code. However, I noticed that [3] has publicly released both code and model weights on GitHub. Including a comparison with [3] would make the evaluation more complete and convincing.

4) Video Comparisons with Open-Sourced Methods: In the supplementary video, there are no comparison demos with existing open-sourced methods such as SadTalker, EchoMimic, and so on. Including video comparisons would provide a clearer understanding of the advantages and limitations of the proposed method and help readers better assess its performance.

References:
[1] Loopy: Taming Audio-Driven Portrait Avatar with Long-Term Motion Dependency
[2] EMO: Emote Portrait Alive -- Generating Expressive Portrait Videos with Audio2Video Diffusion Model under Weak Conditions
[3] Hallo2: Long-Duration and High-Resolution Audio-driven Portrait Image Animation

**Questions For Authors:**

No.

**Relation To Broader Scientific Literature:**

It improves the performance of long-term talking head video generation by achieving robust identity preservation, synchronized expressions, and accurate lip-audio alignment.

**Theoretical Claims:**

Yes.

---

> ### Author Rebuttal · Authors · 2025-04-01
>
> Dear Reviewer Vnrq:
>
> Thank you for your thoughtful and encouraging review. We appreciate your recognition of our work as **“an interesting idea”** with **“rich and comprehensive experiments.”** We have carefully considered all your suggestions. Please find our point-by-point responses below.
>
> **Q1: Ablation Study on Historical Frame Length**
>
> **Response:** Thank you for the insightful question. We conducted an ablation study on the number of archived frames, as shown in Table 8 of the supplementary material, covering 2, 4, 8, and 16 frames. Additionally, we extended the study to 24 frames. Due to memory constraints, further increases were infeasible.
>
> | **Method**   | **FID ↓** | **FVD ↓** | **Sync-C ↑** | **Sync-D ↓** | **SSIM ↑** | **E-FID ↓** |
> |--------------|-----------|-----------|--------------|--------------|------------|-------------|
> | MegActor-Σ   | 48.57     | 724.40    | 7.22         | 7.14         | 0.745      | 2.29        |
> | Ours-2       | 45.72     | 704.26    | 7.39         | 7.03         | 0.754      | 2.19        |
> | Ours-4       | 44.53     | 694.61    | 7.55         | 6.93         | 0.767      | 2.08        |
> | Ours-8       | 43.68     | 678.49    | 7.68         | 6.81         | 0.773      | 2.03        |
> | Ours-16 | 42.08 | 656.71| 7.84    | 6.69     | 0.779  | 1.97 |
> | **Ours-24**  | **41.74** | **642.35**| **7.96**     | **6.55**     | **0.782**  | **1.94** |
>
> The experimental results show that even with only 2 frames, our method outperforms MegActor-Σ across all metrics, indicating strong robustness under limited historical context. Furthermore, increasing the number of archived frames consistently enhances identity preservation and motion quality. We will highlight this ablation more clearly in the revised manuscript.
>
>
> **Q2: Lack of Qualitative Comparison with Related Works [1][2], Recent Open-Sourced Hallo2, and Other Methods**
>
> **Response:** Thank you for the valuable suggestion. We have added qualitative comparisons between our proposed MCDM and both open-source methods (SadTalker, EchoMimic, and Hallo2) and closed-source methods (Loopy [1] and EMO [2]), using the official audio and image inputs provided on the project pages of Loopy and EMO. Please refer to the following demo links: **Loopy**: [Demo 1](https://files.catbox.moe/d54l8i.mov), [Demo 2](https://files.catbox.moe/zezhpy.mov) and **EMO**: [Demo 1](https://files.catbox.moe/77kqif.mov), [Demo 2](https://files.catbox.moe/1p4qo4.mov).
>
> Our method demonstrates stronger identity consistency and better motion synchronization compared to SadTalker, EchoMimic, and Hallo2, while achieving visual quality comparable to Loopy. Additionally, it generates more natural expressions and overall enhanced visual realism compared to the original EMO outputs.
>
>
>
> **Q3: On the Distinction Between GAN-based and Diffusion-based Methods**
>
> **Response:** Thank you for the helpful suggestion. We have revised the introduction to improve the logical flow and clarify the distinction between GAN-based and diffusion-based methods. GANs often suffer from training instability and artifacts such as flickering, especially in long-form generation. In contrast, diffusion models provide more stable training and improved visual realism via multi-step denoising and stronger identity preservation, making them better suited for this task. We have added this clarification and cited relevant works in the revised version.
>
>
> **Q4: On Pose Constraints and Expression Diversity**
>
> **Response:** Thank you for the constructive feedback. (1) Our original intent was to highlight that when the reference image and driving pose come from different identities, strong pose constraints (e.g., facial landmarks) may cause identity distortion or unnatural warping. (2) While facial expressions are generally independent of head pose, over-constraining pose can inadvertently limit expression diversity. For example, methods like EchoMimic use dense 3D landmarks from tools such as MediaPipe, which may implicitly encode facial details and constrain expression variation. We have clarified this point in the revised introduction.
>
>
> **Q5: Misuse of the Term “Static Audio Feature”**
>
> **Response:** Thank you for pointing this out. The term “static audio feature” was a misphrasing. Our intended meaning was to describe the limitation of relying solely on audio features without incorporating complementary cues such as lip or head motion. We have clarified this wording in the revised manuscript.
>
>
> **Q6: Redundancy in Contribution Statements**
>
> **Response:** We agree with the reviewer’s observation. Contributions 2, 3, and 4 all describe technical components of our framework. We have merged them into a single contribution for improved clarity.
>
> **Again, thank you so much for helping us improve the paper!  Please let us know if you have any further questions. We are actively  available until the end of this rebuttal period. Looking forward to  hearing back from you!**

---

> > ### Comment · Reviewer_Vnrq · 2025-04-08
> >
> > Thanks for your detailed responses. I appreciate the additional experiments, qualitative comparisons, and clarifications, which have improved the quality of the paper. Given these improvements, I am happy to maintain my original rating and support the paper's acceptance. However, I also respect the suggestions of the other reviewer and the AC.

---

> > > ### Author Response · Authors · 2025-04-09
> > >
> > > Dear Reviewer Vnrq,
> > >
> > > Thank you very much for your follow-up comment and for maintaining your support for our paper. We sincerely appreciate your recognition of our additional experiments, qualitative comparisons, and clarifications. Your constructive feedback helped us significantly improve the quality and clarity of the paper.
> > >
> > > **Although the author-engaged discussion phase will be over by today, if you have any additional questions or open discussions, please don't be hesitant to leave more comments. We are always available at all time, to actively address any concerns or be prepared for more discussions.**
> > >
> > > **Your opinions are rather important for us to improve the work!**
> > >
> > > Thank you!
> > >
> > > Sincerely,
> > >
> > > The Authors

---

### Decision · Program_Chairs · 2025-05-01

**Decision:**

Accept (poster)

**Comment:**

This work makes contributions in: 1). a ​motion-prior diffusion framework that enhances ID consistency and lip-sync quality, 2). a ​memory-efficient attention mechanism, and 3). ​TalkingFace-Wild, a valuable 200H multilingual dataset. The method is well-validated, and the released data benefits the community. The rebuttal successfully addressed concerns, with two reviewers raising their scores. All three reviewers now recommend acceptance. After reading the whole review process and careful discussion, the AC recommends ​Accept.

Authors are required to:
- Incorporate all new materials from the rebuttal into the paper.
- Guarantee dataset release before the camera-ready due.